# Methods for Detection, Extraction, Purification, and Characterization of Exopolysaccharides of Lactic Acid Bacteria—A Systematic Review

**DOI:** 10.3390/foods13223687

**Published:** 2024-11-19

**Authors:** Manoj Kumar Yadav, Ji Hoon Song, Robie Vasquez, Jae Seung Lee, In Ho Kim, Dae-Kyung Kang

**Affiliations:** Department of Animal Biotechnology, Dankook University, 119 Dandae-ro, Cheonan 31116, Republic of Korea; yadavmk@dankook.ac.kr (M.K.Y.); 72190292@dankook.ac.kr (J.H.S.); rvasquez@dankook.ac.kr (R.V.); dlwotmd2587@dankook.ac.kr (J.S.L.); inhokim@dankook.ac.kr (I.H.K.)

**Keywords:** characterization, exopolysaccharides, lactic acid bacteria, purification, structure

## Abstract

Exopolysaccharides (EPSs) are large-molecular-weight, complex carbohydrate molecules and extracellularly secreted bio-polymers released by many microorganisms, including lactic acid bacteria (LAB). LAB are well known for their ability to produce a wide range of EPSs, which has received major attention. LAB-EPSs have the potential to improve health, and their applications are in the food and pharmaceutical industries. Several methods have been developed and optimized in recent years for producing, extracting, purifying, and characterizing LAB-produced EPSs. The simplest method of evaluating the production of EPSs is to observe morphological features, such as ropy and mucoid appearances of colonies. Ethanol precipitation is widely used to extract the EPSs from the cell-free supernatant and is generally purified using dialysis. The most commonly used method to quantify the carbohydrate content is phenol–sulfuric acid. The structural characteristics of EPSs are identified via Fourier transform infrared, nuclear magnetic resonance, and X-ray diffraction spectroscopy. The molecular weight and composition of monosaccharides are determined through size-exclusion chromatography, thin-layer chromatography, gas chromatography, and high-performance liquid chromatography. The surface morphology of EPSs is observed via scanning electron microscopy and atomic force microscopy, whereas thermal characteristics are determined through thermogravimetry analysis, derivative thermogravimetry, and differential scanning calorimetry. In the present review, we discuss the different existing methods used for the detailed study of LAB-produced EPSs, which provide a comprehensive guide on LAB-EPS preparation, critically evaluating methods, addressing knowledge gaps and key challenges, and offering solutions to enhance reproducibility, scalability, and support for both research and industrial applications.

## 1. Introduction

Polysaccharides are a kind of long-chain polymer mainly produced by animals, plants, and microbes with various biological functions, and those excreted outside the cell wall are known as exopolysaccharides (EPSs) [1,2]. EPSs are a group of large-molecular-weight bio-polymers released during the metabolic pathways of many microbes, such as fungi and bacteria, into their surrounding environment [3]. Sutherland coined the word EPS in 1972 [4] to characterize a variety of microbial polysaccharides found outside of cell walls, which can be loosely or tightly attached according to their structural and functional interactions with the cell [5]. EPS protects bacterial cells from extreme temperatures, salinity, osmotic pressure, and desiccation [6]. Bacteria are most widely utilized for the production of EPSs due to their quick multiplication ability and weakly attached EPS with the mucoid surface, which can be simply isolated from cells using any isolation method [7]. EPSs can be divided into two forms, such as homo-exopolysaccharides (HoEPSs) and hetero-exopolysaccharides (HeEPSs), based on their monosaccharide compositions [8]. HoEPSs comprise similar building blocks, such as monosaccharides, primarily glucose or fructose, whereas HeEPSs are typically branched and comprise multiple building blocks, including glucose, mannose, xylose, fructose, arabinose, and rhamnose [5,9,10]. The EPSs are produced by microbes either by adhering to the cell surface, creating a capsule, or by releasing it into the surrounding medium during the growth and metabolism [11,12]. EPSs are nontoxic, biocompatible, biodegradable, insoluble in all other polar solvents, and soluble in water, indicating the presence of hydrophilic residues in the polymeric structure [7,13].

Furthermore, every strain of bacteria releases various kinds of EPSs along with various biological functions and can be used alone or mixed with other compounds for a broad range of biomedical and pharmaceutical applications [14]. The quantity, chemical makeup, and biological activity of EPSs can be influenced by factors, such as the producing strain, media composition, and culture conditions [15,16]. To produce new EPSs with considerable biological activity, it is necessary to carefully select EPS-producing strains, optimize production conditions, and provide precise information on the EPS structure [17].

Several lactic acid bacteria (LAB) are broadly capable of producing distinct kinds of EPSs consisting of repeating units of sugars or sugar derivatives [18,19,20]. The LAB-producing EPS has received extensive interest because of its lack of toxicity and biodegradability [2,8]. LAB produces EPSs in two forms, such as a layer of mucous, which forms a colloidal aggregation as a translucent, mucilaginous material surrounding the cell, with little or loose cell adhesion via non-covalent interactions, and a capsule form, which becomes strongly adherent and attaches to the cell surface via covalent bonds [21,22]. EPS-producing LAB have several important advantages for applications in food industries, such as increasing the flavor and rheological characteristics of fermented food and bread crumbs, as well as increasing the softness of bakery products [10]. LAB-EPSs often contain various monosaccharides (glucose, galactose, rhamnose, and mannose) and functional groups that can give rise to unique structural characteristics, such as branching patterns, that enhance their emulsifying, gel-formation, and thickening properties making them suitable for applications in food, pharmaceuticals, cosmetics, and biotechnology industries [23]. The EPSs produced by LAB have several biological functions, like anti-microbial, anti-biofilm, anti-oxidant, anti-viral, anti-obesity, anti-cancer, anti-inflammatory, anti-diabetic, anti-aging, synbiotic, prebiotic, emulsification, edible-film-coating, capacity to hold water and oil, flocculation, hydrogel, immune-enhancing, gelling, thickening, anti-syneresis, and wound healing, making them attractive for future research and applications in food, pharmaceuticals, and biotechnology [2,8]. The LAB-EPSs are generally regarded as safe, so they have been widely used for EPS production. On the contrary, EPSs from pathogenic bacteria, while beneficial for the bacteria survival may contribute to diseases by promoting chronic infections, resistance to antibiotics, and persistence in the human body through biofilm formation [16].

However, the prior EPS review publications primarily emphasized the EPS monosaccharide composition, molecular weight, structure, and antioxidant and rheological characteristics [24]. The structure, function, applications, yield optimization, biological activities, and applications of LAB-EPSs are provided by Yang et al. [25]. The pathways that contribute to LAB-EPS anti-oxidant, anti-viral, anti-cancer, and immunomodulating action are explored by Saadat et al. [26]. Zhou et al. [27] provide a full description of biosynthetic processes, mechanisms of antioxidant activity, and modifications to LAB-derived EPSs and also explore their gene–structure–function relationship. EPS production by LAB is challenging due to strain-dependent yields, structural variability, and specific environmental requirements. Identifying high-yield strains requires extensive screening, and optimizing production often involves costly fermentation or genetic modification. The diversity in EPS structures and functions necessitates a detailed analysis for consistent quality, while isolating EPSs from other metabolites adds complexity. These challenges highlight the need for advanced, cost-effective methods to make LAB-derived EPSs viable for industrial use. A large variety of techniques have been used for the study of EPSs, but a comprehensive compilation of analytical methods has not been adopted to date. In this review, we provide a comprehensive and in-depth examination of different methods for the detection, extraction, purification, and characterization of EPSs released by several LAB strains.

## 2. Production of Exopolysaccharides

Before extraction, EPS-producing LAB must be grown under optimal conditions because the quantity and distribution of EPSs are highly dependent on growth conditions [6]. The production of EPSs is the result of a biosynthetic pathway, not an energy production pathway, and is closely related to the general reactions of glycolysis [28]. The natural EPS produced by LAB is much lower than the production of EPS by non-dairy bacteria [29]. Recently, several types of techniques have been developed to increase the production of targeted EPSs, including metabolic engineering, co-culture, optimization of culture conditions, and abiotic stresses [30].

### 2.1. Metabolic Engineering

The metabolic engineering approaches for the enhancement of EPS production by LAB have primarily focused on sugar nucleotide precursor synthesis [30]. In this method, the genes from the central metabolic pathway are overexpressed, which are known to provide the sugar nucleotide precursors, such as the uridine diphosphate glucose (UDP-glucose) and thymidine diphosphate glucose (dTDP-glucose) required for EPS biosynthesis [31]. The conversion of the glycolytic intermediate glucose-6-phosphate (G6P) to glucose-1-phosphate (G1P) by the enzyme phosphoglucomutase (PGM) and the synthesis of UDP-glucose and dTDP-glucose from G1P catalyzed by UDP-glucose pyrophosphorylase and dTDP-glucose pyrophosphorylase, respectively, are controlling points for the overproduction of HeEPSs [29]. These sugar nucleotides form EPS-repeating unit catalyzed by the EPS gene cluster, such as glycosyltransferases, and assembled on a C55-isoprenoid-lipid carrier molecule, which is attached to the cytoplasmic membrane of the cell, and are other controlling points for the overproduction of HeEPSs [31]. The repeating units are flipped across the membrane via an enzyme flippase (Wzx) encoded by the gene *wzx* and polymerization by WZY polymerase encoded by the gene *wzy* located at the cytoplasmic membrane, which is also a trigger point to enhance the production of HeEPSs [23,31]. On the contrary, HoEPSs are synthesized by only one enzyme, glycosylhydrolase, which catalyzes the polymerization of HoEPSs, and it is easy to enhance the production by changing the enzyme [15]. In general, metabolic engineering provides an alternative strategy for the maximum production of EPSs. One of the difficulties with this strategy is the complexity and diversity of EPS biosynthesis pathways, which can be strongly dependent on the bacterial strains.

### 2.2. Co-Culture

Co-culturing microbes has been shown to increase their tolerance to environmental changes, enabling them to carry out complex metabolic processes [23,30]. The production of EPS is one of the complex metabolic processes activated by co-culture [31]. The effect of bacterial co-culture has been widely researched in association with inactivated *Saccharomyces cerevisiae* via heat treatment or disruption with glass beads, which promoted the production of EPS [32]. Bertsch et al. [33] suggested that the production of EPS was enhanced 49% by co-culturing the *L. rhamnosus* R0011 with *S. cerevisiae*. Similarly, Yamasaki-Yashiki et al. [32] found that co-culturing *Lactobacillus (L.) kefiranofaciens* JCM 6985 with *S. cerevisiae* IFO 0216 increased EPS production by reducing lactic acid accumulation.

### 2.3. Optimization of Culture Conditions

Several factors affect the production of LAB-EPSs, such as components of the medium (nitrogen source, carbon source, mineral salts, other substances) and culture conditions (initial inoculum, pH, temperature, incubation time, and presence/partial presence/lack of oxygen) [18,34,35]. The carbon sources, such as glucose, sucrose, and lactose, are one of the most significant factors that are used or supplemented with the MRS medium to achieve the maximum level of increased EPS production [14,34]. The high production of EPS can be enhanced not only by carbon sources but also by a high carbon/nitrogen ratio and other nutrients, such as zinc, phosphate, calcium, magnesium, and iron that influence EPS biosynthesis [30]. The de Man, Rogosa, and Sharpe (MRS) medium is the most frequently used medium for the production of EPSs, while other media have also been employed, including Brain Heart Infusion (BHI), Luria–Bertani (LB) broth, Mueller–Hinton broth, M17 media, and Kenner Faecal Streptococcal agar media [14,23]. The pH of the growth medium can also increase the production of EPS, with previous studies showing that neutral pH is beneficial for higher production of EPS [30]. The incubation temperature and time used for the production of EPS are different and vary from strain to strain [14]. Shi et al. [36] achieved 3-fold higher EPS production using *L. fermentum* TDS030603 at pH 6.5 in a chemically defined medium supplemented with 5% glucose and 1% ammonium citrate after incubation at 30 °C for 48 h. Midik et al. [37] found that *L. plantarum* MF460 produced higher EPS (515.48 mg/L) at pH 6.0 and after incubation at 30 °C for 48 h. Angelov et al. [38] used three different carbon sources, glucose, sucrose, and fructose, in MRS for the production of EPS from *Lactiplantibacillus (Lpb.) plantarum* ZE2 at 25 °C for 72 h without shaking (Table 1). Optimizing the growth of bacterial strains and culture conditions through changes in the temperature, pH, and media culture composition can significantly increase the production of EPS worldwide. Finding the ideal culturing conditions is difficult because the production of EPS is significantly depending on the type of producing strains.

### 2.4. Abiotic Stress

The overproduction of EPS can be enhanced by abiotic stress, such as high temperature, drought, and salt stress, etc., because bacterial strains produce EPS as a cellular defense response to reduce the severe effects of abiotic stresses [30]. Grosu-Tudor and Zamfir [65] found the maximum production of EPS by *Leuconostoc (Leuc.) pseudomesenteroides* 406 in the presence of NaCl (5%). Seesuriyachan et al. [66] demonstrated the higher production of EPS by *L. confuses* TISTR 1498 with media supplementation with salt (NaCl 5%) stress conditions at 40 °C. According to the criteria mentioned above, the production or overproduction of EPS is a phenomenon that is highly dependent on strain selection, genetic improvement to process control, the medium, and process optimization, which maintain high-quality production in a sustainable and cost-efficient manner [15,34].

## 3. Detection of Exopolysaccharide-Producing LAB Strains

### 3.1. Colony Morphological Analysis of EPS-Producing Cells

Visually examining the morphological characteristics of colonies is the simplest method for the evaluation of the production of EPS [15]. In this method, the LAB strains are screened for the production of EPS based on their colony formation on MRS agar medium supplemented with or without different carbon sources (glucose, fructose, sucrose, galactose, and lactose) under the appropriate incubation conditions [9]. The EPS production is assumed when the colonies of LAB strains are observed in the form of mucoid or slime and ropy [9,54]. The names of different phenotypes, such as mucoid and ropy, are confusing because not all mucoid or slime-producing strains are ropy [6]. Mucoid colonies exhibit a shining and slimy appearance but are unable to develop filaments, although long filaments form from ropey colonies after being touched with an inoculating loop [9,15]. Kavitake et al. [50] used this method to screen the galactan EPS-producing *Weissella confusa* KR780676 and found the slimy colonies on MRS agar supplemented with 2% sucrose. Zaghloul et al. [54] also used a similar method for the initial screening of EPS-producing *Enterococcus* sp. BE11 and found colonies with a mucoid ruby appearance (Table 1). Al-Nabulsi and co-workers confirmed the EPS production by *L. bulgaricus* after the screening of mucoid colonies [9]. Ramos et al. [67] used similar phenotypic methods to screen 123 LAB strains for EPS production and found that 76 LAB strains were able to produce EPS with 33 strains of a ropy and 43 strains of a mucoid phenotype. Colony morphological analysis is the most commonly used method for the initial screening of EPS production because it is simple and cost-effective and can easily examine the phenotypic characteristics of colonies, such as mucoid or slimy and ropy phenotypes, without using expensive equipment. A schematic diagram of different steps involved in the screening of exopolysaccharides of LAB is presented in Figure 1.

### 3.2. Colorimetric Assay for EPS-Producing LAB Strains

#### 3.2.1. Congo Red Agar

The Congo red agar (CRA) is a qualitative analysis used to screen and identify microorganisms capable of producing EPS, particularly used for the staining of a biofilm made due to the EPS [68]. The Congo red (CR) analysis is also used to detect the conformational structure of the EPS, with details mentioned in the characterization section [69]. In this method, the bacterial strains were streaked on the CRA plate and incubated under respective conditions [68]. The CRA typically consists of a nutrient medium (BHI or that suitable for bacterial strains) that provides essential nutrients for microbial growth, sucrose, or glucose and serves as a carbon source to induce EPS production and CR (acidic dye), which binds to polysaccharides, providing a visual indication of EPS production [18]. The production of EPS by microorganisms is confirmed after the visualization of black- or red-color colonies with a dry crystalline consistency in the presence of sucrose, whereas whitish colonies in the presence of glucose and non-producers have transparent colonies [68]. Murugu and Narayanan [18] used the CRA method to detect EPS in the biofilm form produced by *L. amylovorus* MTCC 8129 and found dark red colonies with a dry and crystalline consistency, which confirmed EPS production. Vazquez-Vargas et al. [12] used the CRA method with sucrose and glucose for the screening of the production of EPS from 13 different native LAB belonging to the genera *Lactococcus* and *Lactiplantibacillus* and found black- and white-color colonies on culture medium with sucrose and glucose, respectively, which confirmed the production of EPS (Table 1).

#### 3.2.2. Ruthenium Red Agar Method

The ruthenium red agar (RRA) method is used to screen the production of EPS from ropy colonies and also distinguish the ropy and non-ropy colonies of LAB strains [15,70]. Ruthenium red stains the cell wall of bacteria and produces pink/red-color colonies for non-ropy strains, whereas it appears as white colonies for ropy strains [56,70]. The appearance of white-color colonies is due to the presence of EPS, which prevents the uptake of this stain [71]. In this method, the filter-sterilized ruthenium red dye (0.8% *w*/*v*) is added to the MRS agar medium, streaked/inoculated (on paper disk) with an overnight-grown culture on an assay plate, and incubated for 48 h at 37 °C or optimized conditions [72]. Prete et al. [71] used a ruthenium red stain to screen the ropy (white color) and no-ropy (pink color) LAB strains and found that *Lpb. plantarum* O13, C9O4, and LT100 showed white-color colonies, which confirmed the production of EPS, whereas *Lpb. plantarum* LT53 showed pink-color colonies and was considered a non-EPS producer. Rahnama Vosough et al. [56] used a similar method to determine the production of EPS from 79 different strains of *Enterococcus* species and found that all strains showed EPS production (Table 1). Kamigaki and Ogawa [73] detected slime EPS production from the cell wall of *L. helveticus* SBT2171 using the ruthenium red agar method.

The colorimetric assay can be used as a confirmatory test for the EPS-producing strains because Congo red binds to EPS and produces a dark red color, whereas ruthenium red does not bind to ropy colonies due to the EPS and can be applied for the differentiation of ropy and non-ropy strains.

### 3.3. Microscopic Visualization of EPS-Producing Cells

#### 3.3.1. Light Microscopy

Light microscopy, in combination with bacterial cell staining, provides an easy, quick, and low-cost technology that can be used to identify bacteria that produce EPS [74]. Crystal violet (CV) staining is used to observe the EPS produced by LAB strains under the light microscope [22]. CV is a non-specific dye that binds cellular substances, such as peptidoglycans found in both dead and live cells, and the extracellular matrix, such as nucleic acids, proteins, lipids, and EPS [11]. In this method, the EPS-producing LAB strains are grown in MRS broth medium with or without carbon supplements, like sucrose, under suitable conditions. The EPS-producing culture of LAB is spread on a glass slide and allowed to dry in the air [46]. The dried LAB culture is stained with CV (1% *v*/*v*) for 2 min, washed with copper sulfate solution (20% *w*/*v*), dried in air, and examined under a light microscope [22]. Luang-In et al. [75] used a similar method to visualize the EPS around the cells of *Bacillus tequilensis* PS21 isolated from milk kefir. Nachtigall and co-workers observed that the light areas around the cells of *Streptococcus thermophilus* indicate capsular EPS under a light microscope after staining with CV [46]. Lee et al. [22] also used a similar CV staining method to determine the production of EPS by *Lacticaseibacillus* (*Lcb.) paracasei* DA-BACS and found purple-color-stained EPS around the cells, which suggested the capsular structure of EPS.

#### 3.3.2. Scanning Electron Microscopy

The EPS production by LAB strains around the cells is also identified using a scanning electron microscope [22]. In this method, the overnight-grown cell pellets are washed thrice with phosphate-buffer saline (pH 7.4) or sodium cacodylate (0.1 M) and fixed in a mixture of glutaraldehyde with paraformaldehyde in the same buffer for 45 min or overnight, and the fixative is removed using the same buffer [34,46]. The sample is stained with ruthenium tetroxide (RuO_4_) for 60 min, washed with double-distilled water (ddH_2_O) to remove extra RuO_4_, freeze-dried and attached to aluminum specimen holders, and coated with carbon (40 nm). The cells are visualized under a scanning electron microscope with a high vacuum at 10 kV beam energy [46]. Kavitake et al. [50] used a similar method and verified the production of galactan EPS from *W. confusa* KR780676 (Table 1). Lee et al. [22] observed the lumps of EPS around the producer cells under a scanning electron microscope, which confirmed the EPS production by *Lcb. paracasei* DA-BACS.

Microscopic analysis can be used to determine whether EPS is tightly attached to the cell surface via covalent bonds forming the capsular polysaccharide or loosely attached to the cell via non-covalent interactions.

## 4. Methods for Extraction of EPS

After production, it is important to separate and purify the EPS from impurities, such as medium contaminants, microbial biomass, nucleic acids (DNA and RNA), and proteins [76]. Several steps are involved in the extraction of EPSs, which may vary depending on the source of microorganisms, types of EPS, complexity of production media, and desired purity [14,15]. Here, two different methods are explained for the extraction of EPSs, such as precipitation and ultrafiltration as mentioned in the Figure 1.

### 4.1. Precipitation

The extraction of EPS using the precipitation method involves several steps, such as biomass removal, protein degradation, and precipitation [76]. In this method, centrifugation is the first step used to remove the cell biomass and collect the cell-free supernatant (CFS) containing EPS [10]. After that, the CFS is filtered with a membrane filter (0.45 μm) [54]. Some studies state that the grown culture is heated for 10-15 min at 90–100 °C to kill the bacterial cells and inactivate the enzymes, and cells are removed using centrifugation [10,77].

The protein precipitation and degradation is the second step to remove the protein and obtain pure EPS [76]. Trichloroacetic acid (TCA) is most commonly used for the precipitation and denaturation of proteins and is applied at concentrations between 2 and 80% (*w*/*v*) at 4 °C for 2 h or overnight [19,38]. TCA dehydrates the hydration shells (layer of water molecules) around proteins leading to precipitation, while others suggested that due to its acidic property, TCA alters the protein structure and causes precipitation, although the exact mechanism of precipitation is not clear [78]. Other reagents are also used for protein precipitation and degradation, such as Sevag reagent made from a mixture of chloroform:n-butanol (4:1) and protease enzyme hydrolysis (50 μg/mL) used to eliminate the protein contamination from EPSs [79,80].

The precipitation is the third-most important step used to precipitate EPS using one, two, three, or four volumes of ice-cooled 95 or 100% ethanol (*v*/*v*) and incubation overnight at 4 °C [14,42,63]. Ethanol disrupts the hydration shell created by water molecules interacting with the hydrophilic portions of EPS because it is less polar than water, reducing the EPS solubility [81]. Other precipitating agents, such as propanol, acetone, and isopropanol, are sometimes applied to precipitate EPSs [18,82]. The participated EPS is collected using centrifugation, dissolved in ddH_2_O or Milli-Q water, and purified using different methods, as mentioned in the next section [83]. Yu et al. [10] precipitated the EPS from the TCA (80% *w*/*v*)-treated CFS of *Lpb. plantarum* HDC-01 using three volumes of ice-cold 95% ethanol (*v*/*v*) at 4 °C for overnight. Al-Nabulsi et al. [9] used two volumes of cooled ethanol (100% *v*/*v*) to extract the EPS-L from the CFS of *L. bulgaricus* and then deactivated the enzymes and proteins using 20% TCA (*w*/*v*) (Table 1). Rahnama Vosough et al. [79] precipitated the EPS from CFSs of three different strains of *Enterococcus* using three volumes of 96% cold ethanol (*v*/*v*) and removed the proteins through treatment with Sevag reagent.

The precipitation method is widely used due to its simplicity, efficiency, minimal chemical contamination, and the ability to handle large volumes, making it suitable for the extraction of EPS, whereas some disadvantages of this method include incomplete precipitation, the co-precipitation of impurities (proteins, nucleic acids, and other biopolymers), and the large volumes of ethanol required.

### 4.2. Ultrafiltration

Ultrafiltration is often proposed to offer a more rapid and precise approach to separating EPS in place of TCA and ethanol treatments [76,84]. The principle of ultrafiltration is based on size exclusion and pressure-driven separation [85]. Ultrafiltration effectively separates EPSs from smaller impurities, such as salts and small organic molecules, based on size [86]. Molecules larger than the pore size are retained by the membrane (retentate), while smaller molecules and solvents pass through the membrane (permeate) [87]. In this method, the coagulated proteins and cell fractions from the liquid medium are extracted using centrifugation and the EPS using ultrafiltration [44,76]. Firstly, the CFS is filtered using 0.20 μm, and then, ultrafiltration with a membrane with a 10–500 kDa-molecular-weight cut-off (MWCO) is typically suitable, depending on the size of the EPS molecules, and the protein is degraded using TCA [88,89]. The concentrated EPS can be further processed, such as drying, freeze-drying, or re-dissolving in a desired buffer or water, for subsequent analyses or applications. Ziadi et al. [44] extracted the EPS from the CFS of *Lactococcus (Lact.) lactis* SLT10, *Leu. mesenteroides* B3, and *L. plantarum* C7 using ultrafiltration through 10 kDa-cut-off cellulose membranes (Table 1). Yu et al. [90] used a 100 kDa MWCO membrane to extract the EPS from *W. cibaria* 27 (W27). Macedo et al. [91] extracted the EPS from *L. rhamnosus* RW-9595M using ultrafiltration through a 30 kDa MWCO membrane. Donnarumma et al. [92] concentrated the supernatant of *L. crispatus* L1 9-fold using ultrafiltration and obtained an 85% recovery yield.

Ultrafiltration is a membrane-based separation process that utilizes semipermeable membranes to concentrate and purify EPSs, which provides an efficient, scalable, and chemical-free method for concentrating and purifying EPSs. However, challenges, such as membrane fouling, high initial costs, and potential limitations in molecular weight selectivity, need to be managed for successful implementation.

## 5. Methods for Purification of EPS

The purification of EPS is a multi-step process that extracts and separates the required EPS from a microbial culture. The methods of purification can differ according to the particular qualities and amount of purity required for EPS applications [30]. Dialysis is the first step of purification and a very popular method for the purification of EPS using an MWCO membrane [14]. Anion-exchange chromatography (AEC) and size-exclusion chromatography (SEC) are involved in the second step of purification, which is necessary to confirm the extreme purity of the EPS [83,93]. These final purifications become more significant if the recovered EPS is applied for characterization instead of quantifying the EPS production [94]. Over the past ten years, many methods, such as hot air drying, microwave drying, vacuum drying, spray drying (SD), vacuum spray freeze-drying (SFD), and vacuum freeze-drying (FD), have been widely employed to dry polysaccharides [41]. According to recent research, polysaccharides dried using FD have higher antioxidant potential [95]. The suspension of the EPS sample is stored at −20 °C for frozen, and pure EPSs are obtained through freeze-drying for 12 h at −80 °C using a vacuum freeze-drier [10,42]. The pure dried EPS is stored at −20 °C and used for additional analyses, including characterization [14]. The freeze-dried EPS looks like a white fluffy solid [10,49]. A schematic diagram of different steps of purification is shown in Figure 1.

### 5.1. Dialysis

Dialysis is performed as the final stage before drying and is an important procedure because it helps to remove carbohydrates that have a small molecular weight [76]. The recovered precipitates of EPS are dissolved in ddH_2_O or Milli-Q water before dialysis [15,71]. The dialysis is carried out against the ddH_2_O/Milli-Q water or phosphate buffer saline (PBS, pH 7.4) to eliminate the small neutral sugars, media contaminants, small proteins, salts, and damaged portions of the EPS through a dialysis membrane using an MWCO ranging in size, at 6, 8, 12, and 14 kDa, for 2 days with three water changes per day [15,31]. Subsequently, the dialyzed EPS sample is concentrated using a freeze-drier, yielding a white color and the soft and spongy appearance of pure EPS [67]. In the present stage, the estimated EPS yield is indicated by the weight of the obtained EPS. Al-Nabulsi et al. [9] purified the EPS-L samples from *L. bulgaricus* through dialysis using a 20 kDa MWCO membrane against ddH_2_O and then freeze-dried and stored it at −20 °C (Table 1). Tiwari et al. [96] used a 12–14 kDa-MWCO dialysis membrane to dialyze the EPS extracted from *E. hirae* OL616073 against ddH_2_O at 4 °C for two days. Dialysis does not involve harsh chemicals or extreme conditions, efficiently removes small contaminants (salts, residual sugars, solvents, and small metabolites), does not require organic solvents, is simple, and is inexpensive in small-scale or lab environments, but its slow process, potential sample loss, and large buffer requirements make it less ideal for large-scale industrial production without further optimization or complementary purification techniques.

### 5.2. Anion Exchange Chromatography

Anion exchange chromatography is applied to eliminate negatively charged molecules, such as protein/nucleic acid, that attach to positively charged resins, like ion exchange Sepharose, DEAE-cellulose-52, DEAE-Sepharose, and phenyl-Sepharose [14,84,97]. In this method, the dialyzed and lyophilized crude EPS samples are dissolved in Tris-HCl (0.02 M, pH 7.4) buffer and loaded onto the anion exchange columns; subsequently, the gradient elution with Tris-HCl buffer and NaCl (0–0.3 M) with a flow rate of 1 or 2 mL/min is performed [55,58]. Sheng et al. [42] purified the EPS extracted from *L. pantheris* TCP102 using AEC through a DEAE-Sepharose fast-flow column and found three different EPSs, which were further purified using SEC. Jiang and co-workers purified a dialyzed EPS-F2 sample using a DEAE-52 column eluted using buffer with NaCl (0–0.5 M) with a flow rate of 2 mL/min and found a major single peak, which suggested a single EPS [55]. Li et al. [58] used AEC to purify the dialyzed EPS sample through the DEAE-52 column eluted with NaCl (0.1–0.3 M) solution at a flow rate of 1 mL/min and found three peaks, which suggested three different types of EPSs, such as EPS-1, EPS-2, and EPS-3 (Table 1). AEC is a powerful and selective method for purifying charged EPSs from complex mixtures, offering high resolution with potential fractionation, and it effectively removes contaminants with minimal sample loss. However, it requires careful optimization, is unsuitable for neutral or weakly charged EPS, can be expensive, and is highly sensitive to pH and ionic strength making it less ideal for large-scale production without proper resources and planning.

### 5.3. Size-Exclusion Chromatography or Gel-Filtration Chromatography

Size-exclusion chromatography (SEC) or gel-filtration chromatography (GFC) is a very frequently used method for obtaining pure EPS components because it separates EPSs based on their size and molecular weight [84,93]. Several resins are available for SEC, including Sepharose, Sephadex, Sephacryl, Superdex, Seralose, Superose, etc. [98]. In previous research, Sephadex G-75, G-100, Superdex G-200, Sephacryl S-400, and Seralose 6B were utilized to purify EPSs from LAB strains [10,14,93]. In this method, the dialyzed and AEC-eluted fractions of EPS are loaded in the GFC column, and the elution is performed using ultrapure water with an optimized flow rate [55,99]. Yu et al. [10] purified dialyzed EPS using GFC through a Sephadex G-100 and performed the elution with ddH_2_O with a flow rate of 2 mL/min and found a single symmetric peak, which indicated that the isolated EPS was a homogeneous EPS with high purity. Sharma et al. [43] also used GFC to purify the dialyzed EPS through a Sephadex G-75 and eluted it with ddH_2_O at a 1.5 mL/min flow rate and found a single peak of purified EPS. Similarly, Jiang et al. [55] purified EPS-F2 via GFC using a Sephacryl S-400 HR column eluted with ultrapure water at a 1 mL/min flow rate and found one symmetrical peak, which indicated that the sample was homogeneous (Table 1). The liquid EPS is concentrated using a vacuum freeze-drier at −80 °C, and the amount of carbohydrate content is quantified using the following methods. GFC is size-based separation, removing small impurities, with desalting, in a non-denaturing way to purify EPSs, with no need for a special buffer, and no sample loss due to charge. However, it is limited in resolution for similarly sized contaminants, can be slow and resource-intensive, and may not be the best option for large-scale, high-throughput processes unless optimized carefully.

## 6. Methods for Quantification of EPS

The quantitative measurement of EPS is performed using different methods, such as gravimetric, colorimetric, chromatographic, microscopic, and near-infrared spectroscopy [76,94]. Gravimetric is the simplest approach for quantifying EPS production because it involves weighing the powder; however, this method of measurement is inaccurate because any impurities will be present in the final dry weight [15,23]. Pintado et al. [74] used the gravimetric method to determine the dry mass of EPSs isolated from different LAB strains, which were cultivated in different media, and found that the dry weight of EPS ranges from 1068 to 4736 mg/L.

### 6.1. Colorimetric Methods

Colorimetric methods are simple and inexpensive methods used for determining EPS production by quantifying their carbohydrate contents [83]. The carbohydrate content is most commonly measured using the colorimetric phenol-sulfuric acid method [23,100]. Another colorimetric method for the quantification of the carbohydrate content in EPS is the Anthrone-sulfuric acid method [30]. Colorimetric assays are the cheapest and simplest methods used for quantification, but they are not free from interferences that must be considered when quantitative information is required. However, these colorimetric methods determine the amount of total carbohydrate present in the growth medium, which may interfere with the qualification of EPSs.

#### 6.1.1. Phenol-Sulfuric Acid Method

The simplest way to quantify is to weigh the powder, but this approach is inaccurate since any impurities will be included in the final dry weight [76]. The most commonly used colorimetric method for quantifying the EPS is the phenol-sulfuric acid method, which is faster and more accurate [15,71]. In 1956, Dubois first proposed the phenol-sulfuric acid method [48]. The phenol-sulfuric acid method’s basic principle is that polysaccharides are dehydrated by concentrated sulfuric acid to produce uronic acid and hydroxyurea formaldehyde, which condense with phenol to create orange-red substances [73,100]. In this method, the EPS sample (1 mg) is dissolved in ddH_2_O (1 mL), phenol (50% *v*/*v*, 1 mL) is added, and after that, quickly added concentrated sulfuric acid (5 mL) and glucose, used as a standard [52,74]. The reaction mixture tube is left to stand for 10 min at room temperature (RT), gently shaken, and incubated in the water bath at 25 °C for 20 min [19]. The absorbance of the unique yellow-orange color is measured at 490 nm using a spectrophotometer to determine the glucose equivalent [19,71]. Pintado et al. [74] used this method to quantify the EPSs extracted from different LAB strains cultured in different growth media and found that the EPS concentration ranged from 194 to 1187 mg of EPS/g of dry mass. Prete et al. [71] used the same method to determine the amount of released EPSs during milk fermentation by *Lpb. plantarum* C9O4 and LT100 and found that the strain C9O4 produced 115.55 mg/L of EPS, whereas strain LT100 produced 587.77 mg/L of EPS. The phenol-sulfuric acid method is the most frequently used method for the quantification of EPS because of its accuracy, simplicity, stability, and sensitivity. However, it has several disadvantages, such as non-specificity, interference from other compounds, and sensitivity to experimental conditions, which can affect the accuracy and reliability of the results and need careful consideration when employing this method for EPS analysis.

#### 6.1.2. Anthrone-Sulfuric Acid Method

The detection and quantification of carbohydrates in a solution is determined using another colorimetric technique known as the Anthrone-sulfuric acid method [30,76]. The total quantity of carbohydrates in an EPS sample is estimated using an Anthrone reagent, a mixture of 0.1% Anthrone (*w*/*v*) and concentrated sulfuric acid (80% *v*/*v*) [101]. The EPS is hydrolyzed by sulfuric acid into its monosaccharide components, which condense and form chromophores, hydroxymethylfurfural or furfural derivatives, which react with the Anthrone reagent to form anthranol and produce a blue-green color [78,102]. In this method, the EPS sample (0.5 mL) is mixed with Anthrone reagent (2.5 mL), mixed well, and incubated for 15 min in boiling water (90 °C) [103]. The mixture is chilled on ice for 15 min without exposure to light, and the absorbance at 620 nm is determined [101]. The EPS sample turns into a blue blue-green color due to the presence of carbohydrates [102,103]. Donnarumma et al. [92] used the Anthrone-sulfuric acid method to quantify the EPS isolated from *L. crispatus* L1 and found that the range of EPS content in fermented broth ranged from 200 to 400 mg/L. Behare et al. [104] also used a similar method to quantify the EPSs isolated from *L. fermentum* V10 and *L. delbrueckii* subsp. *bulgaricus* NCDC 285 and found 247.4 and 219.6 mg/L EPS, respectively. Marimuthu and Rajendran [105] used a similar method to calculate the total amount of EPS isolated from *Bacillus subtilis* EPS003 and found the higher yield of 0.939 g/L. The stability of the color reagent is a significant issue with the Anthrone method. Because of the oxidation process that results in the loss of chromomeric characteristics, active anthranol becomes unstable in a strongly protonated solution. Since the Anthrone-sulfuric acid method may only be determined accurately for quantities over 10 mg/L, it is typically used for strains with higher productivity.

### 6.2. Chromatographic Methods

In the chromatographic method, the SEC used to accurately determine the EPS concentration and the matching elution peak can be quantified using refractive index measurements with a refractive index detector (RID) [23,94]. Wolter et al. [106] determined the production of EPSs by *W. cibaria* MG1 and found a maximum 4.2 g EPS/kg of sourdough prepared from buckwheat using SEC through the Superdex 200 column. Prasanna et al. [107] also used SEC to determine the yield of EPS in yoghurt produced by *Bifidobacterium longum* subsp. *infantis* CCUG52486 and found the EPS content of 150.4 mg/kg. The EPS can also be quantified using liquid chromatography techniques, like AEC, high-performance liquid chromatography (HPLC) combined with RID, and high-performance anion-exchange chromatography pulse amperometric detection (HPAEC-PAD) [76]. Further details of each of these methods are explained in the characterization section. The chromatographic methods provide more specificity and accuracy for the quantification of EPS; however, there are several disadvantages, including complexity, sensitivity to operational conditions, cost, and potential issues with specificity and resolution.

### 6.3. Microscopic Methods

In addition, certain microscopic techniques can be utilized for qualitative, as well as quantitative analysis, such as electron microscopy (EM) and confocal laser scanning microscopy (CLSM) after staining with ruthenium red and fluorescent lectin, respectively [76,83,94]. The identification and dispersion of fluorescent-labeled EPS in food substances is commonly investigated via CLSM [15]. Wei et al. [108] used CLSM and SEM to determine the microstructure and particle size of a CAS-EPS-1 complex purified from *Limosilactobacillus fermentum* A51 and found that the EPS-1 effectively filled the three-dimensional network structure of the casein (CAS) clusters and improved the textural properties of yogurt. On the other hand, electron microscopy (EM) needs a high vacuum and dehydrated samples and is effective in determining structural characteristics as described in more detail in the characterization section.

### 6.4. Near-Infrared Spectroscopy

Near-infrared spectroscopy (NIRS) can be used to rapidly and simultaneously quantify EPSs during fermentation without a pre-isolation method [23,76]. Macedo et al. [109] used this method to quantify EPS production during batch cultures of *L. rhamnosus* RW-9595 M in complex whey permeate medium and found a coefficient of relation of 91%, recommending NIRS as a quick approach for detecting EPSs during the fermentation process. NIRS has some disadvantages, such as limited specificity, dependence on sample preparation, and the need for robust calibration models, which can hinder its effectiveness.

## 7. Methods for Characterization of EPS

Currently, the connection between EPS structure and its functional characteristics could be clearer. Thus, it becomes necessary to characterize the chemical and structural characteristics, like functional groups, monosaccharide composition, kinds of sugar subunits, ring conformation, morphology, molecular weight, degree of branching, and types of glycosidic linkages of LAB-EPS, before investigating its functional characteristics [2,14,110]. Several methodologies are used to analyze and recognize the molecular weight, structure, and composition of EPS; however, not one method is capable of determining each of these characteristics, and therefore, using a combination of techniques is usually necessary [15]. The chemical and physical properties of EPSs are characterized using different types of techniques, such as spectroscopy, chromatography, microscopy, thermogravimetry, differential scanning calorimetry, polarimetry, zeta potential (ZP), and particle size distribution (PSD) [3,111]. A schematic diagram of different steps of characterization is shown in Figure 2.

### 7.1. Analysis of Purity of EPS

UV-Vis spectroscopy is used to characterize the physical characteristics of EPSs to ensure that the sample is free of protein and nucleic acid contamination [2,14,93]. In this method, the crude or purified EPS sample is dissolved in distilled water, and the spectrum between the wavelengths of 190 and 800 nm is measured [103]. UV-Vis spectroscopy is used to determine the purity of EPS extracted from *Lpb. plantarum* HDC-01 by scanning at wavelengths of 200–400 nm [10]. Ali et al. [52] also analyzed the EPS-84B isolated from *E. faecalis* 84B using a UV-Vis spectrometer for trace amounts of proteins and nucleic acids at 260 and 280 nm, and no absorbance was found, which indicated the absence of these molecules. Xiao et al. [41] used UV-Vis spectroscopy to determine the purity of EPS and found a few small peaks at 260–280 nm, which confirm the small quantity of nucleic acids and proteins present in EPS isolated from *L. helveticus* MB2-1 after scanning at a wavelength from 190 to 500 nm (Table 1). Gangalla et al. [112] scanned the EPS at 200–800 nm and found a strong absorption peak at 264 nm, suggesting the presence of polysaccharides. The UV-Vis method can be used to determine whether the EPS is contaminated with other compounds but cannot differentiate between EPS and other carbohydrates.

### 7.2. Analysis of the Structure of EPS

The conformational structure of EPS is detected via Congo red analysis [69]. To investigate the usage of EPS in various sectors, it is necessary to understand its properties, such as the types of sugar subunits, glycosidic connections, and functional groups [14]. Fourier transform infrared (FT-IR) spectroscopy is used for the analysis of functional groups, whereas nuclear magnetic resonance (NMR) spectroscopy is used to investigate the proton and carbon profiling, as well as carbon positioning and glycosidic linkages present among the EPS subunits [52]. X-ray diffraction (XRD) spectroscopy is commonly used to analyze the structures, such as crystallinity or amorphousness, of the EPS [108].

#### 7.2.1. Congo Red Test

Congo red is used to analyze the helix-coil transition and determine the conformational structure of the purified EPS [41,113]. It is believed that the polysaccharides are present in a three-dimensional organized structure, usually a triple-stranded helix shape, and that they combine with Congo red to form a complex in diluted solutions of NaOH ranging from 0.05 to 1.0 M [41,114]. Congo red can form a complex with triple-helix EPSs, as indicated by a bathochromic change in the visible absorption maxima of the spectrum [69,115]. Therefore, a polysaccharide having a triple-stranded helical structure will show a change in the UV-Vis absorption spectrum, while other structures should not show any changes [116]. In this method, the EPS sample (2 mg/mL) is mixed with an equal volume of Congo red solution (80 μmol/L), and various volumes of NaOH solution (1 mol/L) are added to obtain a gradient of final concentrations (0–0.5 M), and it is incubated for 5 min at RT [51,115]. The maximum absorption wavelength of the mixtures is measured at RT (25 °C) using a UV-Vis spectrophotometer at a wavelength range of 200–800 nm [41,69]. Congo red was used to demonstrate complex formation in the c-EPS produced by *L. helveticus* MB2-1, which validated the triple-strand helical shape, according to Li et al. [114]. The EPS purified from *L. sakei* L3 formed a complex with Congo red and showed a random coil structure rather than a triple-stranded helical conformation [115]. Zhao et al. [51] used a similar method to determine the helical structure of XG-3 dextran released by *W. confusa* XG-3 and found a random-coil chain structure, rather than a triple helical structure (Table 1). The random-coil structure creates a more open and flexible structure, which improves the solubility of EPS in water. This enhanced solubility makes it easier for the EPS to trap water molecules, forming a hydrated gel-like structure that can protect bacteria from desiccation and help maintain a stable microenvironment.

#### 7.2.2. Fourier Transform Infrared (FT-IR) Spectroscopy

The distribution of the functional groups in the pure EPS is determined using FT-IR spectroscopy, which also analyzes the chemical structure because the structure determines the function of EPSs [108,117,118]. In this method, the freeze-dried EPS sample and dry potassium bromide (KBr) are mixed well at a ratio of 1:100, crushed finely, and subsequently compressed into a pellet with a thickness and diameter of 1 and 10 mm, respectively [3,10]. Attenuated total reflectance (ATR) is used to record the FT-IR spectra of the EPS at a resolution of 4 cm^−1^ in the 4000–400 cm^−1^ range [9,38,45]. All EPSs show carbohydrate peaks via FT-IR spectroscopy, such as stretching vibrations, which are shown to peak at around 3200–3600 cm^−1^ for hydroxyl groups (O-H), 2800–3000 cm^−1^ for C-H groups, 1600–1800 cm^−1^ for carbonyl groups (C=O), 1400–1450 cm^−1^ (symmetric) and 1600–1650 cm^−1^ (asymmetric) for carboxylate groups, 1000–1200 cm^−1^ for glycosidic linkages, and 800–900 cm^−1^ for the pyranose form of glucose [19,35,40,119]. Wei et al. (2023) [108] used FT-TR spectroscopy and did not find an absorption peak at 1730 cm^−1^ and suggested that the EPS-1 purified from *Limosilactobacillus fermentum* A51 was neutral and not an acidic EPS. Xu et al. [34] determined the chemical structure of the EPS purified from *L. casei* NA-2 using FT-IR spectroscopy, and the EPS revealed specific bands corresponding to carbonylated and hydroxylated polysaccharides. Angelov et al. [38] used FT-IR spectroscopy for the determination of structural characteristics of EPS-1 and EPS-2 purified from *Lpb. plantarum* ZE2 and suggested that there were no differences in the glycosidic bonds or sugar rings of both EPSs. Yu et al. [10] analyzed the functional group composition of the EPS purified from *Lpb. plantarum* HDC-01 using FT-IR spectroscopy and showed that the EPS was composed of glucose/glucopyranose subunits linked via an α-(1 → 6) glycosidic bond and contained an α-(1 → 3) branching structure (Table 1). Typically, multiple hydroxyl groups present in the EPS structure help in scavenging activity based on their ability to donate electrons or hydrogen upon interactions with free radicals. The glycosidic linkage has a significant impact on the bioactivity of EPS regardless of other physicochemical properties. The scavenging activity of EPS has positive effects via α-(1 → 6) glycosidic linkages because these linkages can provide flexible regions and increase the accessibility of reactive groups for their interactions with free radicals.

#### 7.2.3. Nuclear Magnetic Resonance (NMR) Spectroscopy

Nuclear magnetic resonance (NMR) spectroscopy is a powerful analytical technique widely used to determine chemical molecular structures, such as the carbon and proton profile, along with the carbon arrangement and configuration of glycosidic bonds found within the EPS [14,15]. The NMR spectroscopy can be divided into two types, such as One-dimensional (1D) (proton (^1^H), carbon-13 (^13^C), nitrogen-15 (^15^N), and phosphorus-31 (^31^P) etc.) and Two-dimensional (2D) (correlation spectroscopy (COSY), heteronuclear single quantum coherence (HSQC), heteronuclear multiple bond correlation (HMBC), nuclear overhauser effect spectroscopy (NOESY) and total correlation spectroscopy (TOCSY)), with each offering different capabilities and applications [120,121]. The chemical structure of EPSs is examined using 1D NMR spectroscopy and also used to determine the detailed information about the hydrogen and carbon environments in the molecule [120]. 2D NMR spectroscopy is used to determine the detailed chemical structure of the EPS through COSY, HSQC, HMBC, NOESY, and TOCSY [60,114].

In this method, the purified dried EPS sample (20 mg/mL) is dissolved in deuterium oxide (D_2_O) and 1D NMR (^1^H-NMR), ^13^C-NMR, and 2D NMR (NOESY, COSY, HMBC, TOCSY, and HSQC) are analyzed at 25 or 70 °C using a spectrometer equipped with a cryo-probe at 400 MHz for H-NMR and 100 MHz for C-NMR [114]. Chemical shifts (*δ*) are reported in parts per million (ppm) [121]. Ayyash et al. [60] used 1D (^1^H and ^13^C) and 2D (^1^H-^1^H TOCSY, ^13^C-^1^H HSQC, and HMBC) NMR spectroscopy to determine the chemical structure of EPS-M41 purified from *Pediococcus pentosaceus* M41 and proposed the structure as → 3)α-D-Glc(1 → 2)β-D-Man(1 → 2)α-D-Glc(1 → 6)α-D-Glc(1 → 4)α-D-Glc(1 → 4)α-D-Gal(1 → with arabinose linked at the terminals. Al-Nabulsi et al. [9] also examined the structural composition of EPS-L purified from *L. bulgaricus* using 1D and 2D NMR spectroscopy and suggested that the presence of uronate functions likely in either the α- or β-anomeric configuration. Similarly, Yu et al. [10] used 1D (^1^H and ^13^C) and 2D (COSY, NOESY, HMBC, and HSQC) NMR to investigate the chemical structure and bond configuration within the EPS purified from *Lpb. plantarum* HDC-01 and suggested that the EPS was a homopolysaccharide linked via an α-D-(1 → 6) glycosidic bond and contained an α-(1 → 3) branching structure (Table 1). The NMR results revealed the presence of α and β anomeric sugars and an α (1 → 6) glycosidic linkage, which confirmed the prominent role of EPS as an antioxidant compound. The anti-tumor activity of EPS is closely associated with the presence of β-(1 → 3) glycosidic bonds in the EPS and β-(1 → 6) glycosidic bonds in the branches of the EPS.

#### 7.2.4. X-Ray Diffraction (XRD) Spectroscopy

XRD spectroscopy is used to evaluate the crystalline and amorphous characteristics of EPS using an X-ray diffractometer, which helps in understanding the physical characteristics of EPSs [14,40]. The crystal-structured EPS are more likely to extend and form a rigid conformation in the food system, which increases the binding of EPS to the food system, resulting in increased viscosity of the food system and also affects the other physical properties of the EPS, such as emulsification, swelling capacity, and solubility [108]. XRD uses X-rays to determine the geometry or shape of compounds, and two basic principles contribute to X-ray diffraction from crystalline compounds [122,123]. First, the elastic scattering of photons applies the laws of specular reflection and treats the atomic planes as mirrors. Second, crystals diffract X-rays because their wavelength is similar to the inter-atomic distances (0.15–0.5 nm) in the crystals, which lead to constructive and destructive interference phenomena [122]. In this method, the purified EPSs are uniformly mounted on a quartz carrier, and its crystalline structure or diffractogram is analyzed between a range of two-theta (2θ) angles (5–90°) at a scanning speed of 2 or 10° per minute at RT [10,61]. The angle of diffraction is changed to identify any changes in the crystal structure [40]. Du et al. [49] used XRD within a spectrum of 2θ angles (5–80°) using a scanning speed of 10° per minute and found the non-crystalline amorphous nature of H2 dextran purified from *W. confusa* H2. Yu et al. [10] also used XRD to determine the non-crystalline amorphous nature of the EPS purified from *Lpb. plantarum* HDC-01. Du and co-workers used XRD and found the non-crystalline amorphous nature of the EPS purified from *Leuc. pseudomesenteroides* [64]. Similarly, Ge et al. [40] used XRD to determine the crystallinity of EPS and found the amorphous nature (Table 1).

### 7.3. Analysis of Molecular Weight of EPS

The molecular weight (MW) of EPS is determined using gel permeation chromatography (GPC) or SEC and high-performance size-exclusion chromatography (HPSEC) with RID and multi-angle laser light scattering (MALLS) detectors [48,72]. The physicochemical characteristics and biological activity of LAB-EPS are significantly influenced by its MW and determined using GPC or SEC and HPSEC [15,110]. EPS functions are significantly affected by their MW; a lower MW is associated with the bioactivity (antioxidant activities) of compounds, whereas a higher MW increases the viscosity in aqueous solutions and exhibits potent anti-cancer activity [52]. The lower MW of an EPS has been directly related to anti-oxidant activity because a small EPS complex can also easily penetrate the cell and protect against the free radical. In this method, the purified dry EPS (1 or 2 mg) is dissolved in 1 mL sodium nitrate (NaNO_3_, 0.1 M) for 1 h at 60 °C, cooled, and filter-sterilized using a cellulose membrane filter (0.2 or 0.45 μm) [10]. The stationary phase consists of small-size beads with small pores (Waters Ultra hydrogel linear or Ohpak SB-804 HQ column), while the mobile phase is a liquid (0.1 M, NaNO_3_), and the fractions are eluted with a flow rate of 0.5 or 1 mL/min [3,64]. The eluted fractions are detected with RID and MALLS detectors at 35 °C combined in HPSEC or an RID detector combined with GPC/SEC, whereas the column temperature is between 40 and 45 °C [45,124]. Different molecular weights of different monosaccharides, such as glucose, dextran, and pullulan, are used as standards, and a standard curve is prepared based on the retention time of the elution peak [3,124]. Yu et al. [10] used an Ohpak SB-804 HQ analytical column with RID and MALLS detectors and found the MW 2.505 × 10^6^ Da that was purified from *Lpb. plantarum* HDC-01. Ayyash et al. [45] also used an RID and pullulan and confirmed that the MW of EPS-C70 was 3.8 × 10^5^ Da, which was purified from *L. plantarum* C70. Similarly, Du and co-workers used Ohpak SB-804 HQ analytical columns with RID and MALLS detectors and found the MW of HDE-9EPS to be >1.0 × 10^6^ Da purified from *Lvlb. brevis* HDE-9 [61] (Table 1).

### 7.4. Analysis of Monosaccharides Composition of EPSs

The structures of EPSs produced by various probiotic LAB strains are distinct because the composition of monosaccharides depends on the media composition, type of strains, and culture conditions, like water activity, pH, and temperature [83]. The composition of monosaccharides is an important feature affecting the biological activities of EPSs [108]. Normally all monosaccharides occur in the D-configuration except for L-arabinose, L-fucose, or L-rhamnose, which become essential components that can influence the biological activity of EPSs, such as anti-tumor, anti-inflammatory response, anti-oxidant, etc. [125]. The monosaccharide composition of an EPS influences the viscosity of food; in particular, when the amount of glucose in the monosaccharide composition is increased, its thickening capacity is greatly strengthened, which is advantageous for improving the viscosity of food, whereas the type of glycosidic bonds of an EPS significantly influence the bioactivity, such as anti-oxidant and immune activities [108]. Therefore, the composition of monosaccharides and glycosidic bonds of EPSs is important. The purified EPSs are hydrolyzed using enzymes or acids, and the released monomers are determined using different chromatographic methods, such as thin-layer chromatography (TLC) or high-performance thin-layer chromatography (HPTLC), HPLC combined with UV or RID or diode array (DA) detectors, gas chromatography (GC), gas chromatography-mass spectrometry (GC-MS), and HPAEC-PAD [15,46,72,97].

#### 7.4.1. TLC or HPTLC

TLC is used to qualitatively determine the monosaccharide composition of an EPS [126]. The principle of TLC is the movement of a substance between a fixed solid phase applied on a glass or aluminum/plastic plate and a liquid mobile phase that moves over the solid phase [127]. In this method, the purified EPS (10 mg) is acid hydrolyzed along with standard carbohydrates, such as glucose, arabinose, sucrose, galactose, fructose, maltose, glycerol, galacturonic acid, glucuronic acid, and xylose using trifluoroacetic acid (TFA, 2 or 6 M) or sulfuric acid (2 N) at 100 °C for 2 or 6 h [47,126]. The majority of carbohydrates are hydrolyzed in the presence of TFA because of their ability to break down the glycosidic bonds without causing severe damage to monosaccharide molecules [112]. In some studies, the TFA is removed using a rotary vacuum evaporator, and the hydrolyzed sample is neutralized with an ammonia solution (15 M) and after that is spotted on the TLC plate [126,128]. A small amount of hydrolyzed EPS sample (1 or 5 µL) is spotted near the bottom (1–2 cm above from bottom) of the TLC or HPTLC silica gel plate (30 × 20 cm) using a capillary tube/micropipette or HPTLC automatic sample applicator and left to dry [103,126].

The chromatogram is developed in an equilibrated (with Whatmann’s filter paper) TLC chamber using a mobile phase, such as ethyl acetate:n-propanol:acetic acid:H_2_O at a ratio of 4:2:2:1 [47,103,128], ethyl acetate:acetic acid:1-butanol:H_2_O at a ratio of 4:3:2:2 [112], n-butanol:methanol:25% ammonia solution:H_2_O at a ratio of 5:4:2:1 [129], and butanol:ethanol:H_2_O at a ratio of 5:3:2 [126]. The level of mobile solvent should be below the spotted line, cover the chamber, and allow the solvent to rise in the plate through capillary action. Once the solvent front is about 1–2 cm from the top of the TLC plate, remove the TLC plate and dry it at RT. The spots of monosaccharide components are visualized by spraying the developer, such as ninhydrin [103], 0.5% 1-naphthyl ethylenediamine dihydrochloride in methanol with 5% sulfuric acid (*v*/*v*) [112], orcinol-sulfuric acid [47], ethanolic p-anisaldehyde + ethanolic conc. sulfuric acid [126], and aniline-diphenylamine [128], and the plate is dried at 70, 100, or 120 °C for 5, 10, or 15 min according to the developer.

In the HPTLC, the bands are visualized under white light, scanned, and analyzed by the CAMAG visualizer system [126]. Sharma et al. [43] used the TLC method to determine the monosaccharide components and confirmed the existence of glucose, galactose, and mannose in extracted EPS from *L. paraplantarum* KM1 (Table 1). Kwon et al. [129] also used the TLC method to analyze the monosaccharide components and found that the EPS is mainly composed of glucose. Similarly, Adesulu-Dahunsi et al. [130] used n-butanol:ethanol:H_2_O (50:30:20) as the mobile phase and visualized the spots of EPS purified from *W. cibaria* GA44 using anisaldehyde-sulfuric acid as the developer after heating for 30 min at 110 °C and suggested that the EPS is a heteropolysaccharide composed of glucose and rhamnose. TLC is used for the qualitative analysis of the monosaccharide component of an EPS; however, it has low discriminating power.

#### 7.4.2. HPLC

The monosaccharide composition of purified EPS is detected and quantified via HPLC technique in combination with different detectors [117,131]. In this method, the purified EPS is dissolved in TFA (2 M) and subsequently heated for 2 h at 120 °C in a closed glass tube filled with nitrogen (N_2_) to hydrolyze the EPS into monosaccharide components [118,132]. Once hydrolysis is finished, the hydrolyzed sample is cooled at RT, and the supernatant is collected, filter-sterilized (0.2 or 0.45 μm), and dried using methanol under N_2_ steam thrice to completely remove TFA [133]. In some studies, the purified EPS is hydrolyzed with sulfuric acid (1 M) and neutralized with 1 N sodium hydroxide (NaOH) [3]. The hydrolyzed EPS and standard monosaccharide mixture is mixed with a methanolic solution of 1-phenyl-3-methyl-5-pyrazolone (PMP, 0.5 M) and NaOH (0.3 M) for derivatization [131,132]. The mixture is incubated at 70 °C for 60 min to allow the reaction, then cooled at RT, and neutralized with hydrochloric acid (HCl, 0.3 M) [133]. To remove the unreacted excess amount of PMP, chloroform is mixed into the mixture and strongly shaken, and the chloroform layer is discarded, which is repeated thrice. The aqueous layer is filtered through a membrane filter (0.2 or 0.45 μm) and analyzed via HPLC [134].

Sugar standards are used to produce the calibration curves for galactose, glucose, mannose, and monosaccharide measurements [135]. Each sample (10 or 20 μL) is injected at a time in a reversed-phase C18 column and is assisted in the analysis of monosaccharide derivatives using an HPLC system equipped with a detector, such as UV, RID, or DAD [133,136]. The elution is carried out at 24 °C with a 1 mL/min flow rate with the mobile phase. The mobile phase is composed of a gradient system of solvent A and solvent B in which solvent A comprised 1-butylamine (0.2%), phosphoric acid (0.5%), and tetrahydrofuran (1%) in H_2_O, whereas solvent B consisted of acetonitrile (50%) and solvent A (50%) [43]. Xu et al. [34] used HPLC to confirm the monosaccharide composition of EPS extracted from *L. casei* NA-2, which contained rhamnose, glucose, and mannose residues with molar ratios of 24.3:1.0:42.9, indicating that the EPS is a heteropolysaccharide. Yu et al. [10] also used HPLC and found glucose/glucopyranose subunits in the EPS purified from *Lpb. plantarum* HDC-01 (Table 1). Similarly, Vijayalakshmi et al. [131] also used HPLC to discover the monosaccharide components, such as glucose, rhamnose, and galactose, in EPS partially purified from *Leu. mesenteroides* 201607 and suggested that it is a heteropolysaccharide EPS. Wei et al. [108] used HPLC to detect the monosaccharide component of EPS-1 of *Limosilactobacillus fermentum* A51 and found fucose, which plays an important role in its anti-inflammatory, anti-aging, and moisturizing effects.

#### 7.4.3. High-Performance Anion-Exchange Chromatography with Pulsed Amperometric Detection (HPAEC-PAD)

The monosaccharide composition of purified EPS is analyzed using HPAEC-PAD [14,117]. HPAEC is an effective analytical technique for carbohydrate differentiation because it can distinguish all types of amino-sugars, alditols, mono-, oligo-, and polysaccharides based on structural characteristics, such as the composition, size, linkage isomerism, and anomericity [137]. In this method, the hydrolyzed EPS sample and standards (25 μL), as mentioned in the above section, are sterilized with a membrane filter (0.2 μm) and injected into an HPAEC-PAD system equipped with CarboPac PA1 and CarboPac PA10 columns (normally used for monosaccharide separation) to determine the monosaccharide composition [59,138]. The elution is performed with NaOH (2 or 18 mM) at a 0.45 mL/min flow rate, followed by a linear gradient between 0 to 1 M sodium acetate in 200 mM NaOH at 30 °C to elute acidic monosaccharides [39,138]. The temperature of the column and detector compartment is 30 and 25 °C, respectively [124]. Zhu et al. [39] employed HPAEC-PAD equipped with a CarboPac PA20 column and discovered that EPS-1 isolated from *L. curvatus* SJTUF 62116 was largely constituted of glucose and mannose with a relative molar ratio of 1:1.05. Similarly, Jiang et al. [59] used HPAEC-PAD and found that the EPS-E8 purified from *P. pentosaceus* E8 primarily comprises glucose, mannose, and galactose at a molar ratio of 18.12:80.39:1.49 (Table 1).

#### 7.4.4. Gas Chromatography (GC)

GC is an analytical technique used to detect, identify, and quantify the types of monosaccharides (simple sugars) present in EPSs after acetylated derivatives [62,97,139]. The monosaccharide analysis is mainly involved three steps, such as the hydrolysis, reduction of aldehyde groups, and acetylation. The EPSs are complex polysaccharides and need to be hydrolyzed into simple monosaccharides, such as aldose and uronic acid, with TFA, as mentioned above. The monosaccharides are not volatile and cannot be directly analyzed via GC, so they need to be chemically modified in a process called derivatization. Before the reduction, the hydrolysis of lactone is an essential step because some uronic acids exists in a lactone form, for which the aldehyde group cannot be reduced by reducing agents, such as sodium borohydride (NaBH_4_) or sodium borodeuteride (NaBD_4_) [140]. The lactone is converted to sodium uronate by mixing sodium carbonate (Na_2_CO_3_, 0.5 M) with hydrolysate at 30 °C for 45 min in the presence of ammonium hydroxide (NH_4_OH, 0.5 M) [141]. NaBH_4_ is added to the mixture with ddH_2_O to reduce the aldehyde group in aldose and uronic acid and convert it into alditol and aldonic acid, respectively [140,142]. The reaction is terminated by adding glacial acetic acid (25% *v*/*v*), drop by drop, until no bubbles are observed [63]

The extra glacial acetic acid is removed using a rotary evaporator at 60 °C, and the mixture is loaded on a cation-exchange column and eluted with ddH_2_O to remove the sodium (Na^+^) ions. The eluted fractions are dried via evaporation, and the borate ions (BO_3_^3−^) in the form of methyl borate are removed by adding the methanol four times [140]. The methanol is evaporated at 60 °C, and the sample is heated at 85 °C for 4 h to completely convert the aldonic acids into aldonolactones because aldonic acid cannot react with n-propylamine [141]. The sample is dissolved in a mixture of pyridine and n-propylamine to convert aldonolactones into n-propylaldonamide for 30 min at 55 °C and dried via evaporation (60 °C) [119]. The pyridine and acetic anhydride (CH_3_CO)_2_O) (1:1) are added to the sample and heated for 90 min at 80 °C to convert alditol and n-propylaldonamide into alditol acetate and n-propylaldonamide acetate, respectively [140,141]. The extra pyridine and acetate anhydride are removed from the sample via evaporation at 80 °C and reconstituted in methanol or chloroform [63].

The monosaccharide composition is analyzed using GC coupled with a flame ionization detector (FID) through columns, such as an HP-5 silica capillary column (30 m × 0.32 mm × 0.25 mm), DB-5MS column, or HP-Ultra-2 (25 m × 0.20 mm × 0.11 μm), and different monosaccharides, such as glucose, xylose, galactose, arabinose, and mannose, are used as standards [9,45,143]. The filter sterilized (0.2 or 0.4 µm) sample (1 µL) is introduced in the split-less injection mode using helium or N_2_ as a carrier gas at a 1.5 mL/min flow rate. The alditol acetate derivatives are separated using the following temperature gradient: 80 °C for 2 min, 80–170 °C at 30 °C per min, 170–240 °C at 4 °C per min, 240 °C held for 30 min, and samples are ionized via electron impact at 70 eV [40,144]. Ayyash et al. [45] used GC-FID to determine the monosaccharide analysis of EPS-C70 and found that the major monosaccharide constituents were arabinose, mannose, glucose, and galactose with a molar ratio of 2.7:1.4:15.1:1.0. Similarly, Al-Nabulsi et al. [9] also used GC-FID to determine the monosaccharide composition of EPS-L purified from *L. bulgaricus* and found three central monosaccharide units, such as glucose, mannose, and galactose with a molar ratio of 14.2:1.2:0.75 (Table 1).

#### 7.4.5. Gas Chromatography-Mass Spectrometry (GC-MS)

The methylation analysis is a more advanced technique used to determine the glycosidic linkages within the EPS via premethylation, hydrolysis, reduction, acetylation, and GC-MS analysis [117]. The type and location of glycosidic bonds in an EPS are significant for certain biological activities, such as the reductive power associated with arabinose (1 → 4) and mannose (1 → 2) linkages and radical-scavenging ability associated with glucose (1 → 6) and arabinose (1 → 4) linkages. The immunosuppressive and immunostimulatory activities of mannose-predominant EPSs are primarily attributed to the existence of β-(1 → 2) and β-(1 → 6) linkages [125]. In this method, the purified EPS sample is dried in a vacuum oven at 40 °C overnight and completely dissolved in dimethyl sulfoxide using sonication and mixed with sodium hydroxide (NaOH) for 3 h in the presence of N_2_ at RT [134]. The premethylation reaction is started using methyl iodide (CH_3_I) for 180 min, and the reaction is stopped through the addition of ddH_2_O, and the methylation is confirmed via FTIR [39]. The methylated EPS is extracted with dichloromethane (CH_2_Cl_2_), washed thrice with ddH_2_O, and dried under N_2_ steam to obtain methyl derivatives [145]. Subsequently, the methylated EPS is converted into partially methylated alditol acetates (PMAAs) after hydrolysis with TFA, reduction with NaBD_4_, and acetylation with (CH_3_CO)_2_O as mentioned above [39,134]. The PMAAs are dried under N_2_ and re-dissolved in acetone and injected into the GC-MS system for analysis [145]. Glycosidic linkage analysis identification is performed based on the matching degree of the PMAA mass spectra from the Complex Carbohydrate Research Center spectral database for PMAAs [39]. Wu et al. [146] used GC-MS to determine the chemical composition of EPS purified from *Liquorilactobacillus mali* T6-52 (formerly known as *Lactobacillus mali*) and found that the EPS was primarily composed of glucose. Zaghloul et al. [54] also used GC-MS and suggested that the BE11-EPS was made up of different monosaccharide moieties: galactose, rhamnose, glucose, arabinose sugar derivatives, and glucuronic acid. Similarly, Zhu et al. [39] used GC-MS to determine the methylation analysis of EPS-1 purified from *L. curvatus* SJTUF 62116 and found that the EPS was primarily composed of glucose and mannose at a molar ratio of 1:1.05 (Table 1).

#### 7.4.6. Polarimetry

Polarimetry is one method used to determine the optical configuration of an EPS [14]. The optical configuration of a monosaccharide is based on the direction of rotation of plane-polarized light. The sugar rotates plane-polarized light to the right (clockwise) and is known as the D-configuration or dextrorotatory, whereas the sugar rotates light to the left (counterclockwise) known as the L-configuration or levorotatory. Bajaj et al. [147] used a polarimetry study and presented a positive value of optical rotation revealing the dextrorotatory nature of EPS of *E. faecium*.

### 7.5. Analysis of Morphological Characteristics

The microstructure and surface morphology of EPSs are important for certain functions, such as the linear structure having high viscosity and good water solubility, irregular and rough surface indicating attraction towards water molecules, and pseudoplastic behavior [108,139]. The microstructure and surface morphology of the purified EPS are analyzed using atomic force microscopy (AFM) and a scanning electron microscope (SEM), which are considered powerful tools and used to determine the physical properties of purified EPSs [30,136,144].

#### 7.5.1. Atomic Force Microscope (AFM)

An AFM is a powerful photographic method that is used to determine the surface and topographical patterns of purified EPS [133,148]. An AFM is a type of scanning probe microscope, for which the near-field method is based on the contact between atoms of the sample surface and a sharp tip or sensor, which is attached to a very flexible cantilever [149]. The cantilever systems work in three open-loop modes, such as the non-contact, contact, and tapping mode [150]. In this method, the purified EPS sample is thoroughly dissolved (1 mg/mL) in ddH_2_O under an N_2_O gas flow and agitated continuously for 1 h [144]. The EPS solution (0.01 mg/mL) is uniformly distributed on a clean mica sheet and dried at RT or with N_2_ gas flow [139]. The molecular morphology and three-dimensional (3D) structure of the EPS samples are afterward observed using an AFM in tapping mode with cantilever oscillation [136,151]. Jiang et al. [59] determined the topographical AFM images of EPS-E8 purified from *P. pentosaceus* E8 and found spherical clusters suggesting the molecular aggregation of polysaccharide chains. Yu et al. [10] detected an irregular surface with bumps of varying lengths and some irregular compound when they utilized an AFM to analyze the surface morphology of EPSs isolated from *Lpb. plantarum* HDC-01 and suggested that the EPS samples had good water-holding capacity and biocompatibility and could be widely used in the pharmaceutical field (Table 1). Similarly, Zhao et al. [139] observed round lumps and chains on the irregular and rough surfaces of the EPS purified from *Leu. lactis* L2, which indicated the affinity to water molecules and pseudoplastic behavior.

#### 7.5.2. Scanning Electron Microscope (SEM)

An SEM is used to examine the morphological properties of purified EPS, such as the surface morphology and distribution of molecules [83,118]. SEM equipment operates on the concept of impinging a fine beam of high-energy electrons on the surface of the sample and collecting a diversity of signals from the surface of the sample to evaluate its characteristics [152]. In this method, the freeze-dried EPS samples (2–20 mg) are mounted on a SEM stub, gold-sputtered (2–30 nm thick), and examined using an SEM under a voltage of 3 or 20 kV at 100× and 25,000× magnifications [10,16,63,69]. EPS has a characteristic surface morphology and is therefore related to normal dextran [144]. Ge et al. [40] determined the surface morphology of EPS extracted from *L. delbrueckii* subsp. *bulgaricus* using an SEM after being fixed on a metal stub and coated by a layer of gold and found an irregular porous with stacked flakes appearance on the EPS surface. Similarly, Derdak et al. [63] used an SEM to determine the morphological analysis and found smooth and lotus leaf shape of EPS-SL purified from *Leu. mesenteroides* SL, while EPS-N5 purified from *E. viikkiensis* N5 showed a stiff-like, porous appearance, and compact structure. Jiang et al. [59] also used an SEM to determine the morphological analysis of EPS-E8 purified from *P. pentosaceus* E8 and observed that the spherical structure, rough reticular-like shape, and rough surface can be employed as plasticized biofilm materials (Table 1). The porous structure of an EPS has high water-holding capacity and high solubility in water-based solutions, creating the EPS fast-swelling system, which benefits various applications, such as emulsifiers, stabilizers, and gelling agents in the food industry [40]. The flat and smooth surface of the EPS indicates that the EPS can improve the rheological properties of food and promote the viscosity and water-holding capacity [10].

### 7.6. Analysis of the Elemental Composition

The elemental composition of the pure EPS is accomplished using scanning electron microscope energy dispersive X-ray (SEM-EDX) spectroscopy to investigate the oxygen, carbon, phosphorus, sulfur, and nitrogen compositions [136,153]. The SEM produces more information about the sample when combined with EDX [148]. The weight and atomic percentages of the recorded elements are determined using the X-rays that are released [16]. Zanzen et al. [53] used SEM-EDX and found that carbon (42.43%) and oxygen (39.41%) are the major elements in the EPS purified from *E. mundtii* A2. The EDX analysis of EI6-EPS showed the majority of elements, like oxygen (55.44%) and carbon (40.06%), as well as the existence of nitrogen (2.15%) and phosphorus (0.99%), along with additional elements, such as chloride (1.03%), calcium (0.19%), sodium (0.12%), and magnesium (0.02%), without sulfur [16]. Similarly, Zaghloul et al. [54] applied SEM-EDX to analyze the elemental composition of BE11-EPS purified from *Enterococcus* sp. BE11 and concluded that carbon (49.11%) and oxygen (39.98%) served as the main components, with heteroatoms like phosphorus (2.18%) and nitrogen (6.24%) indicating the presence of protein and phospholipids (Table 1).

### 7.7. Analysis of Thermal Characteristics of EPS

The knowledge of thermodynamic characteristics is helpful for a better understanding when a particular material could have greater possibilities for commercial application significance, thus contributing to the fast production of novel products [62]. The examination of the thermal characteristics contributes to the understanding of the physicochemical characteristics of EPSs and increases their industrial use [10,83]. Thermal behavior, like thermal stability and the melting point of an EPS, is calculated via thermogravimetric analysis (TGA), differential scanning calorimetry (DSC), and derived thermogravimetric (DTG) analysis [14,118]. TGA determined the decomposition/weight loss and DSC measured the endothermic and exothermic changes, whereas DTG analyzed the derivative of weight loss of the EPS sample during the temperature rise [154]. In this method, the purified EPS sample (3 or 10 mg) is placed in an alumina oxide (Al_2_O_3_) crucible, and the thermal properties of the EPS are determined using a thermal analysis instrument. The sample is heated from 40 to 800 °C at a linear heating rate of 10 °C/min under an argon or nitrogen gas with a flow rate of 50 mL/min and initially calibrated for calorimetry and temperature using indium as the melting standard [63,135]. Ayyash et al. [45] used DSC to determine the thermal behavior of EPS-C70 purified from *L. plantarum* C70 and found two endothermic peaks at 158.76 and 76.95 °C equivalent to melting point I and glass transition (Tg), respectively, and one exothermic peak at 247.90 °C, which suggested that EPS-C70 is highly stable during food thermal treatments. Du et al. [61] also used TGA and DSC to analyze thermal properties and suggested that HDE-9 EPS purified from *Lvlb. brevis* HDE-9 showed high thermostability with a degradation temperature (Td) of 302.85 °C, which suggests good thermal stability, making it suitable for application in the food industry. Similarly, Yu et al. [10] used TGA, DSC, and DTG to analyze the thermal nature of EPS purified from *Lpb. plantarum* HDC-01 and found good thermostability, which indicated that the EPS can be used in a wide range of applications in the food, pharmaceutical, and chemical industries (Table 1).

### 7.8. Analysis of Zeta Potential and Particle Size of EPS

The presence of electric charge distribution and stability of EPS is determined through zeta-potential measurements [83,155]. The zeta potential is determined by pouring a solution into a cell, which has two gold electrodes, and when the voltage is supplied to the electrode, particles travel towards the electrode with an opposite charge [156]. A popular tool for characterizing sediment particles is the particle size analyzer, which uses the samples being studied to diffractionally illuminate a laser light source. Based on light diffraction, the analyzer is used to determine the size distribution of a powder, solution, or emulsion [157]. The zeta potential value evaluates the colloidal suspension stability of a solution, and high value is associated with high solubility [57]. The purified EPS is dissolved in the ddH_2_O and sterilized through a membrane filter (0.45 μm), the zeta potential and particle size are determined at RT (20–23 °C), and RI (1.332), viscosity (0.88 cP), and the dielectric constant (78.3) are determined using the zeta potential and a particle size analyzer [9,55]. Bamigbade et al. [57] determined the particle size of 311.2 nm and zeta potential of −12.44 mV of EPS-C15 purified from *Lact. lactis* C15 and suggested that the higher zeta potential values are linked to better solution stability. Jiang et al. [59] suggested that the both the zeta potential absolute value (22.3 to 33.85 mV) and average particle diameter (44.1 to 164.1 nm) increased gradually with an increasing concentration of EPS-E8 purified from *P. pentosaceus* E8. Similarly, Zhao et al. [51] found a zeta potential of 8.6 mV and particle size of 353.2 nm of EPS dextran purified from *W. confusa* XG-3 and suggested that the high absolute values of the zeta potential correspond to high stability (Table 1). The variations in particle size distribution and zeta potential between EPSs produced by different LAB suggested clear differences in their physical and chemical nature.

## 8. Conclusions

EPSs are a fascinating and dynamic area of research with significant implications across various fields, including medicine, biotechnology, agriculture, and environmental science. EPSs produced by LAB have satisfactory advantages over other natural agents in industrial applications, as well as in health promotion and therapeutics. The choice of one or more methods for the EPS will depend on the microbial strain, the culture media, and the degree of accuracy and precision required for subsequent studies. The sucrose is the best carbon source for EPS production in combination with MRS medium, whereas the ropy and mucoid colonies are the initial screening method. The ethanol precipitation is a valuable method for isolating of EPSs due to its simplicity, cost-effectiveness, and scalability. However, it also has limitations related to incomplete precipitation, potential sample loss, and the need for optimization. Dialysis is the most commonly used method for the purification of EPS; however, AEC and GPC are also used depending on the desired purity. The phenol-sulfuric acid method is widely used for the quantification of EPS because it is easy to perform and suitable for a broad range of carbohydrate concentrations. The characterization of EPSs involves a combination of methods to fully understand their chemical structures, physical properties, and functional characteristics. The molecular weight of EPS is widely determined using GPC, and chemical structures are analyzed via FT-IR, NMR, and XRD, whereas the monosaccharide composition and linkage groups are elucidated via TLC, HPLC, and GC. It seems clear that the different types of culture media and methods used to isolate and measure EPSs have a strong influence on the performance results, making it difficult to compare values. The morphological analysis of EPSs is the most commonly determined using an AFM and SEM, whereas the thermal stability of EPS is evaluated via TGA and DSC. In the present study, the principle and purpose of various methods are explained, which helps researchers to understand the detailed characteristics of EPSs and encourage them to discover new/novel EPSs from probiotic LAB. Characterizing the structural properties and biological activities of newly discovered EPSs is essential to expanding their applications. Optimizing microbial strains through genetic engineering, improving fermentation technologies, and exploring cost-effective substrates (such as agro-industrial wastes) could make EPS production more commercially viable. Numerous challenges must be overcome to fully understand the potential of EPSs, such as increasing EPS production, investigating the structure–function relationships, and understanding their particular effects on human health.

## Figures and Tables

**Figure 1 foods-13-03687-f001:**
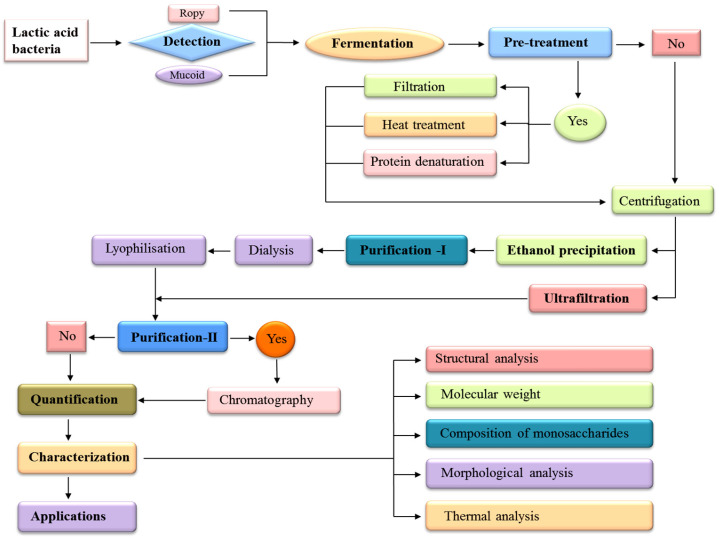
A schematic diagram of various steps involved in the detection, production, extraction, purification, and characterization of exopolysaccharides (EPSs) of lactic acid bacteria (LAB).

**Figure 2 foods-13-03687-f002:**
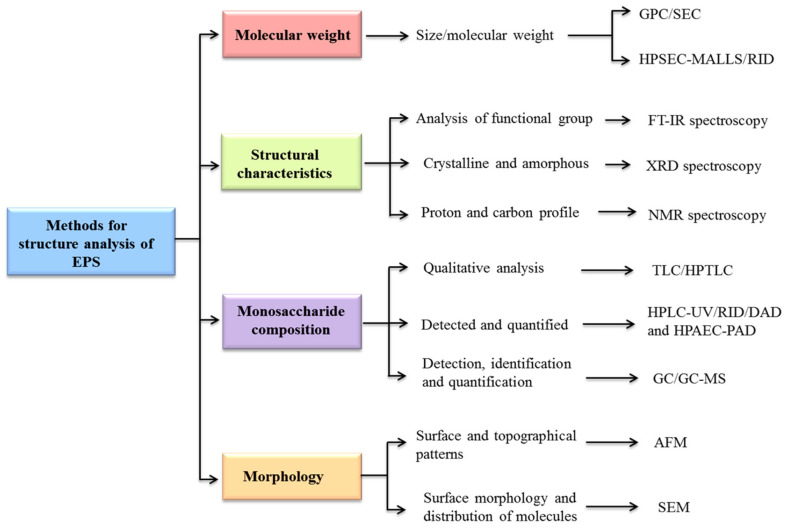
A schematic diagram of various methods used for the determination of different structural characteristics of exopolysaccharides (EPSs) of lactic acid bacteria (LAB).

**Table 1 foods-13-03687-t001:** List of different methods used for the detection, production, extraction, purification, and characterization of exopolysaccharides (EPSs) of lactic acid bacteria (LAB).

Genus	Strains	EPS	Detection	Production	Extraction	Purification	Characterization	References
*Lactobacillus*								
	*L. bulgaricus*	EPS-L	Colony morphology	MRS with sucrose	Precipitation (EtOH) and TCA	Dialysis	GPC, GC-MS, FT-IR, NMR, SEM, and ZP	[9]
	*L. curvatus* SJTUF 62116	EPS-1	-	MRS	Precipitation (EtOH) and TCA	-do-	UV, HPSEC-MALLS-RID, FT-IR, NMR, HPAEC-PAD, GC-MS, SEM, AFM, and TGA	[39]
	*L. delbrueckii* subsp. *bulgaricus* *Strain* IMAU40160	EPS	-	MRS	Precipitation (EtOH) and Sevage reagent	-do-	GC–MS, FT-IR, NMR, XRD, and SEM	[40]
	*L. helveticus* MB2-1	EPS	-	Whey powder, lactose, and soya peptone	Precipitation (EtOH) and TCA	Dialysis, AEC, and GFC	UV-VIS, FT-IR, NMR, HPLC, SEM, DSC, TGA, ZP, and PSD	[41]
	*L. pantheris* TCP102	EPS1	-do-	MRS	-do-	Dialysis, AEC, and GFC	FT-IR, UV, HPLC, and SEM	[42]
	*L. paraplantarum* KM1	EPS	-	MRS with lactose	-do-	Dialysis and GFC	HPLC and SEM	[43]
	*L. plantarum* C7	EPS	-	MRS with sucrose	Precipitation (EtOH) and ultrafiltration	-	FT-IR, GC-FID, and HPSEC-RID-MALLS	[44]
	*L. plantarum* C70 (KX881779)	EPS-C70	-	MRS with sucrose	Precipitation (EtOH)	Dialysis	GPC-RID, GC, FT-IR, NMR, DSC, SEM, ZP, and PSD	[45]
	*L. plantarum* NS1905E	EPS-NS1905E	-	MRS	-do-	-do-	GPC-MALLS, HPAEC-PAD, and FT-IR	[46]
	*L. plantarum* Ts	EPS	Colony morphology	MRS with sucrose	-do-	-do-	FT-IR, TLC, and HPLC	[47]
*Lactiplantibacillus*								
	*Lpb. plantarum* EI6	EPS	Colony morphology	MRS with sucrose	Precipitation (EtOH) and TCA	Dialysis	FT-IR, NMR, GC-MS, SEM, and SEM-EDX	[16]
	*Lpb. plantarum* HDC-01	EPS	-do-	-do-	-do-	Dialysis and GFC	UV, HPLC, GPC, FT-IR, NMR, XRD, SEM, AFM, TGA, DSC, and DTG	[10]
	*Lpb. plantarum* ITD-ZC-107	EPS	CRA	BHI with bagasse and agave	Precipitation (EtOH)	-	TGA and DSC	[12]
	*Lpb. plantarum* Jb21-11	EPS-Jb21-11	Colony morphology	MRS	Precipitation (EtOH)	Dialysis	HPLC-SEC, GC, and NMR	[48]
	*Lpb. plantarum* ZE3	EPS-1 and EPS-2	-	MRS with glucose, sucrose, and fructose	-do-	-do-	UHPLC, HPSEC, and FT-IR	[38]
*Weissella*								
	*W. confusa* H2	H2 EPS	-	MRS with sucrose	-do-	Dialysis and GFC	UV-Vis, GPC, HPLC, FT-IR, NMR, XRD, SEM, AFM, and TGA	[49]
	*W. confusa* KR780676	Galactan EPS	-do-	-do-	Precipitation (EtOH)	-do-	FT-IR, XRD, SEM, and PS	[50]
	*W. confusa* SKP173	EPS	-do-	-do-	-do-	-do-	GC-MS, HPLC, and FT-IR	[19]
	*W. confusa* XG-3	XG-3 EPS	-	Optimized medium with sucrose	-do-	Dialysis and GFC	GC, HPLC, SEM, AFM, FT-IR, XRD, NMR, Congo red, TGA, ZP, and PSD	[51]
*Enterococcus*								
	*E. faecalis* 84B	EPS-84B	-	MRS with sucrose	Precipitation (EtOH) and TCA	-do-	UV, GPC, GC-FID, FT-IR, NMR, SEM, DSC, ZP, and PSD	[52]
	*E. mundtii* A2	EPS	-	MRS	-do-	Dialysis	FT-IR, XRD, SEM-EDX, TGA, and DTG	[53]
	*Enterococcus* sp. BE11	EPS	Colony morphology	MRS with sucrose	-do-	-do-	UV-Vis, FT-IR, NMR, GC-MS, SEM, and SEM-EDX	[54]
	*Enterococcus* sp. F2	EPS-F2	-	-do-	-do-	Dialysis, AEC, and GFC	UV-Vis, HPSEC, HPAEC, FT-IR, GC-MS, NMR, XRD, TGA, SEM, ZP, and PSD	[55]
	*Enterococcus* spp.	EPS	RRA	-do-	Precipitation (EtOH) and Sevag reagent	-do-	HPLC, GPC, and FT-IR	[56]
*Lactococcus*								
	*Lact. lactis* C15	EPS-C15	-	M-17 medium with sucrose	-do-	-do-	GPC, GC-FID, FT-IR, NMR, SEM, DSC, ZP and PSD	[57]
	*Lact. lactis* subsp. *lactis* IMAU11823	EPS-1	-	MRS	-do-	Dialysis and AEC	GC, FT-IR, NMR, SEM, HPLC, ZP, and PSD	[58]
*Pediococcus*								
	*P. pentosaceus* E8	EPS-E8	-	-do-	Precipitation (EtOH) and Sevage reagent	Dialysis, AEC, and GFC	HPSEC-MALLS, HPAEC-PAD, GC-MS, FT-IR, NMR, XRD, TGA, DSC, DTG, SEM, AFM, ZP, and PSD	[59]
	*P. pentosaceus* M41	EPS- M41	-	MRS with sucrose	Precipitation (EtOH) and TCA	-do-	UV, GC-FID, FT-IR, NMR, DSC, SEM, ZP, and PSD	[60]
*Levilactobacillus*								
	*Levilactobacillus brevis* HDE-9	HDE-9 EPS	Colony morphology	-do-	-do-	Dialysis and GFC	UV-Vis, GPC, HPLC, FT-IR, NMR, XRD, SEM, AFM, and TGA	[61]
*Leuconostoc*								
	*Leu.* *mesenteroides* SN-8	EPS-8-2	-	MRS with sucrose	-do-	Dialysis and GFC	UV-Vis, GC-FID, HPSEC, FT-IR, NMR, TGA, and DSC	[62]
	*Leuc. mesenteroides* SL	EPS-SL	Colony morphology	MRS with sucrose	Precipitation (EtOH) and TCA	Dialysis and GFC	UV, FT-IR, NMR, GC-MS, SEM, and TGA	[63]
	*Leuc. pseudomesenteroides*	EPS	-	Glucansucrase with CaCl_2_ and sucrose	-do-	Dialysis	UV, GPC, HPLC, FT-IR, XRD, SEM, AFM, TGA, DSC, and DTG	[64]

## Data Availability

No new data were created or analyzed in this study. Data sharing is not applicable to this article.

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
