# Peer review of "Methods for Detection, Extraction, Purification, and Characterization of Exopolysaccharides of Lactic Acid Bacteria—A Systematic Review"

_foods, 2024, doi:10.3390/foods13223687_

Round 1
Reviewer 1 Report
Comments and Suggestions for Authors
FOODS-3265348
“A review on different methods for detection, extraction, purification, and characterization of exopolysaccharides of lactic acid bacteria” by Manoj Kumar Yadav et al. is meant to be an original review article. However, it is a repetition of earlier publicized reviews about well-known existing methods for detecting, extracting, purifying, and characterizing exopolysaccharides of lactic acid bacteria. One example on this topic is “Lactic Acid Bacteria, Methods and Protocols” by Kanauchi (Editor), Methods in Molecular Biology 1887, Springer Protocols, Humana Press (2019).
Although clearly and systematically described, the present review is an enumeration of methods (with summarized protocols, sometimes literally copied from literature) and too many references to recently published articles (most from East Asian), in which the techniques are used. In this way, a long list of 204 references is created. Some references are unnecessary. Moreover, with a Table occupying 9 pages, the review is getting too comprehensive (35 pages!). A more appropriate selection of examples should be made and revision is necessary, including the following comments.
Line 413: Near-infrared spectrometry must be Near-infrared spectroscopy
Since the properties of LAB-EPS are closely related to their structural characteristics, such as monosaccharide composition, molecular weight, functional groups, glycosidic bonds, and substituents, the techniques to obtain these features (Section 7) need more attention and should be shortened.
Line 489: There are more than two types of NMR spectroscopy (31P, 15N).
The references 86, 119, 127, 130, 135, 136, and 137 are not the most relevant in explaining the use of NMR spectroscopy in the structural analysis of LAB-EPS.
Line 505: Chemical shifts (δ) are always reported in parts per million (ppm) relative to an internal standard (usually tetramethylsilane), not to residual solvent signals.
Line 556: For sugar composition analysis (Section 7.4), it is important to determine the type of monosaccharides (pyranose/furanose) and the D or L configuration of each monosaccharide.
Line 563: -mass spectroscopy must be -mass spectrometry (GC-MS)
Line 589: ninhydrin does not react with hexoses and pentoses, colouring must stem from contamination.
Line 622: What is meant by: .... HPLC system equipped with a RID connecting with the a UV detector detectors.
Section 7.4.4. is unclear, and must be rewritten.
Line 656: -mass spectroscopy must be -mass spectrometry (GC-MS)
The difference between Monosaccharide analysis and Methylation analysis is not clear.
The monosaccharide analysis method is: EPS → Hydrolysis → Reduction (NaBH4) → Acetylation → GC-MS of the Alditol-Acetates
Linkage types must be determined by methylation analysis: EPS → Permethylation → Hydrolysis → Reduction (NaBD4) → Acetylation → GC-MS of the Partial Methylated Alditol-Acetates (PMAAs)
Line 679: A GC-MS can not be coupled to an FID
Line 687: What is hermoph?
Line 691: A GC-MS can not be coupled to an FID
The “Conclusions” contain information belonging to the “Introduction”. Furthermore, the “Conclusions” is a repetition of the enumeration of the discussed methods.
Reviewer 2 Report
Comments and Suggestions for Authors
This article introduces the detection, extraction, purification, and characterization methods of extracellular polysaccharides from lactic acid bacteria, which is very meaningful and has important guiding significance for the research of lactic acid extracellular polysaccharides.
1. In the production of extracellular polysaccharides, the author needs to summarize the production rules, and the summary results can guide actual production.
2. The author needs to introduce the advantages and disadvantages of different extraction methods for EPS for analysis and discussion.
3. The author needs to analyze and discuss the advantages and disadvantages of different purification methods for EPS.
4. There are many quantitative methods for EPS. Why did the author summarize the above methods, and different methods have differences in measuring sugar content? What kind of measurement method is suitable for different EPS?
5. In the analysis of EPS structure, the author needs to summarize the different methods used for different structures and identify patterns in different methods.
6. The author should summarize a key diagram to illustrate the detection methods for studying structural features.
7. Add the practical significance of this study, as well as its prospects and future issues that need to be addressed, at the end of the review.
8. The format of references needs to be standardized, and it is recommended to cite references: DOI: 10.1186/s40538-021-00214-x. 10.1186/s40538-023-00433-4.
Reviewer 3 Report
Comments and Suggestions for Authors
Thank you for the opportunity to participate in the review of the manuscript entitled “A review on different methods for detection, extraction, purification, and characterization of exopolysaccharides of lactic acid bacteria”
The manuscript describes methods for obtaining EPS from lactic acid bacteria.
The manuscript has a typical layout for a review paper. The introduction is written in a very interesting way. It tells the topic well and is supported by the relevant literature. It is very important that the authors cite new publications from recent years. The manuscript is divided into several chapters, which makes it much easier to read and analyze the text. The description itself is very long and detailed, sometimes a bit boring, but it contains a lot of interesting and valuable information. A very important thing in this manuscript is the organization of the text. The analysis of the text is greatly facilitated by the figure and table. The summary of the manuscript refers to the purpose and scope of the work.
In summary, the manuscript is well written and appropriate for Foods.
After taking into account the reviewer's comments and slightly improving the manuscript, it can be forwarded to the next stages of publication.
Below are the reviewer's detailed comments:
My proposal to change the title to a better one: "Methods for detection, extraction, purification, and characterization of exopolysaccharides of lactic acid bacteria - a systematic review”.
Line 29. Keywords. They should be different from the words in the title of the manuscript. This will increase the possibilities of searching for the article in the database.
Line 102. The reviewer would also add presence/partial presence/lack of oxygen
Line 112. The new species name is Lacticaseibacillus casei. Please add it in parentheses.
Line 114. Please provide the full name L. plantarum.
Line 140. Figure 1 should be placed directly below the place (paragraph) where it was cited.
Line 171. The abbreviation L. plantarum was previously used. Please standardize.
Line 583. (v/v/v/v)
Line 623. detector detectors?
Line 626. H2O (subscript)
Line 665. Sodium borohydride explanation is needed.
Line 745. without “and”
Reviewer 4 Report
Comments and Suggestions for Authors
This manuscript summarizes the detection, extraction, purification, and characterization methods of exopolysaccharides (EPS) produced by lactic acid bacteria (LAB). However, these methods and technologies are generally applicable to the extraction and identification of exopolysaccharides extracted from most bacterial strains, and this manuscript lacks in-depth discussion and innovative point. Therefore, it cannot be considered for publication in Foods. Here are some comments:
1. Section 2 introduces the optimal culture media, culture temperatures and time for LAB to produce EPS. This information is quite fundamental and widely covered in existing literature. The authors should provide a more in-depth analysis, such as the response of specific LAB strains to culture conditions, or newly discovered cultivation strategies to enhance EPS production.
2. Sections 4, 5, and 6 introduce the methods for the extraction, purification, and quantification of EPS, which are applicable to the purification, separation, and quantification of exopolysaccharides from most bacteria, not specifically LAB. The authors need to clarify what makes EPS from LAB unique compared to other bacteria and whether special extraction methods are required to highlight these peculiarities.
3. Section 7 extensively demonstrates the information about various techniques and their principles, such as UV-Vis spectroscopy, Congo red test, FT-IR, NMR, XRD, TLC, and HPLC. These technologies help researchers to analyze the structure of LAB EPS, but the authors should focus on the uniqueness of the structure of LAB EPS and discuss in depth the relationship between the structure of LAB EPS and their biological activities and physiochemical characteristics.
4. Although Table 1 lists many types of exopolysaccharides produced by LAB, the authors overlook a critical issue: the EPS yield of these LAB is very low (20-200 mg/L). Is there any special detection or extraction technology that could obtain higher yields of LAB EPS? This is a significant issue that deserves in-depth discussion. The authors should consider the strategies to improve yields or discuss the limitations of current methods and potential improvements.
Round 2
Reviewer 1 Report
Comments and Suggestions for Authors
The manuscript has become a very nice review.
It has sufficiently improved to warrant publication.
Reviewer 4 Report
Comments and Suggestions for Authors
Although authors have revised the manuscript based on my previous suggestions, the revised manuscript is still not recommended for publication. Here are some significant questions, but not limited to these:
(1) The article lacks novelty, merely listing literature. The title of this manuscript is "Methods for detection, extraction, purification, and characterization of exopolysaccharides of lactic acid bacteria". According to the title, the manuscript should focus on the methods and LAB EPS. However, the whole manuscript mainly lists the detection, extraction, purification, and characterization methods applicable to almost all bacteria, without reflecting the uniqueness of LAB EPS. So why not summarize methods applicable to EPS from all bacteria?
(2) Authors have made some modifications in Section 2, but the content is very basic. For example, in Section 2.1 Metabolic engineering, authors merely demonstrate that glucose-1-phosphate is a precursor for the synthesis of various monosaccharides. What is the connection between this and using metabolic engineering to help LAB produce EPS? In Section 2.2 Co-cultivation, how can it be determined that the increase in EPS production is due to the increase in LAB EPS rather than the EPS of newly introduced bacteria? In Section 2.4 Abiotic stress, authors mention that high temperature, drought, and salt stress can promote the excessive production of EPS, but the subsequent examples are all about how carbon sources promote EPS production. What is the difference between that and the content stated in Section 2.3 Optimization of culture conditions?
(3) The authors have not discussed the relationship between the structure of LAB-EPS and their biological activities and physicochemical properties. The content in Section 7 is redundant, and it is necessary to delete it. What are the characteristics of the structure of LAB-EPS, such as monosaccharide composition, molecular weight, and glycosidic bond connection? What activities does LAB-EPS with specific monosaccharide compositions and different molecular weights have?
